# Instance-level Consistent Graph With Unsupervised Human Parts for Person Re-identification

## Abstract

The representation of human parts plays a crucial role in person re-identification (re-ID) by offering discriminative cues, yet it presents challenges such as misalignment, occlusion, and extreme illumination. Previous methods have primarily focused on achieving strict part-level consistency. However, individual part features change inevitably under harsh conditions, hindering consistent representation. In this article, we propose an Instance-level Consistent Graph (ICG) framework to address this issue, which extracts structural information by introducing graph modeling atop unsupervised human parts. Firstly, we introduce an attention-based foreground separation to suppress non-instance noise. Subsequently, an unsupervised clustering method is designed to segment pixel-wise human parts within the foreground, enabling fine-grained part representations. We propose a flexible structure graph that derives instance-level structure from part features, treating each part feature as a node in a graph convolutional network. In essence, ICG mitigates incompleteness through feature flow among nodes, broadening the matching condition from strict part-level consistency to robust instance-level consistency. Extensive experiments on three popular person re-ID datasets demonstrate that ICG surpasses most state-of-the-art methods, exhibiting remarkable improvements over the baseline.

## 1 Introduction

Person re-identification (re-ID) is the task of identifying individuals across different camera views, playing a crucial role in intelligent surveillance systems and finding applications in public security, transportation analysis, smart cities, and more. Recent advancements leveraging human parts extraction (Zhang et al., 2019b; Zhu et al., 2020; Zhang et al., 2021b; Li et al., 2021; Miao et al., 2022) have shown promise by utilizing discriminative part features. However, the detailed information provided by human parts also presents challenges, particularly in terms of part misalignment. Real-world re-ID scenarios often face issues like partial occlusion, pose variations, and extreme illumination (Zhang et al., 2022; Zhao et al., 2022), exacerbating misalignment due to limited information about human parts.

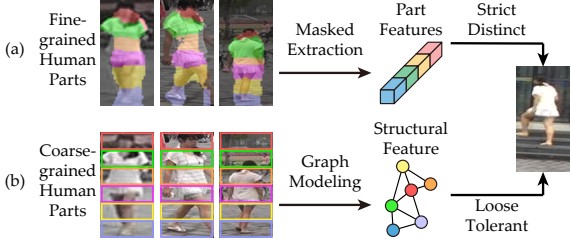

Figure 1: Typical methods against part misalignment. (a) Strictly aligning human parts that requires part-level consistency. (b) Loosely aligning human parts but allows instance-level consistency

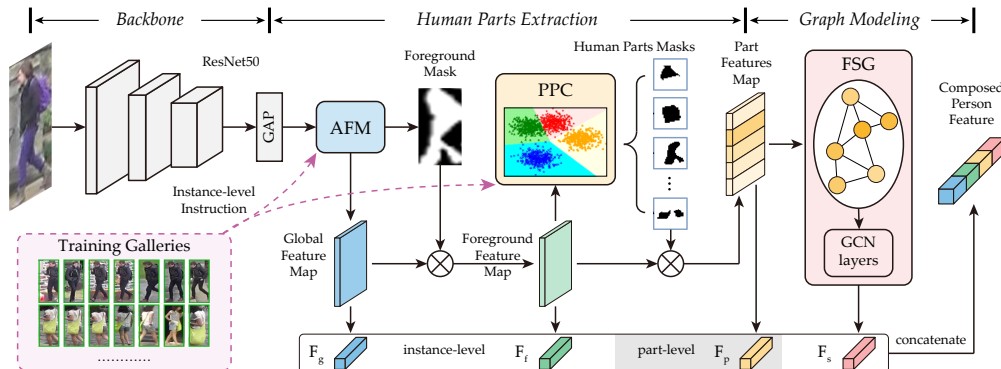

Figure 2: Structure diagram of ICG. Attention-based foreground mask (AFM) and pixel-wise human parts clustering (PPC) are responsible for human parts extraction, flexible structure graph (FSG) is responsible for graph modeling. ResNet50 is adopted as backbone network. AFM and PPC are trained over all samples of each instance rather than over each sample.

To address misalignment, various methods have been proposed to extract more robust part features, which can be broadly categorized into three types: 1) Methods with auxiliary semantics (Su et al., 2017; Kalayeh et al., 2018; Zhang et al., 2019b;a; Miao et al., 2022) introduce additional semantic information such as skeleton pose, human parts segmentation, or bounding boxes, often obtained through pre-trained detection or parsing models. These methods excel in capturing fine-grained part features when precise part localization is achieved. 2) Methods employing local stripes partition (Sun et al., 2018; Zhang et al., 2021a; Yu et al., 2022b) images into horizontal sub-regions, extract local features, and align them vertically. These methods leverage strong positional priors to locate parts effectively, making them simple yet relatively effective in the early stages. 3) Methods leveraging attention mechanisms (Li et al., 2018; Jin et al., 2022; Gong et al., 2022; Jiao et al., 2019) use spatial attention to adjust features at different locations, implicitly aligning part features by emphasizing the foreground and suppressing the background. Despite these efforts, achieving part-level consistency across all samples remains an ideal but stringent assumption, especially under challenging conditions. Extracting identical features from incomplete information about human parts can lead to a semantic gap between samples of the same instance, as briefly illustrated in Fig. 1(a).

Graph convolution networks (GCNs) (Kipf & Welling, 2017) have garnered attention in the field of person re-ID beyond human parts extraction. GCNs offer advantages in topology modeling and learning, making them a focal point in prior works (Shen et al., 2018; Li et al., 2019; Yu et al., 2022a) that utilize GCNs to associate multiple gallery images and probe images. Furthermore, GCNs are well-suited for modeling temporal relations, as evidenced by studies conducted by Bao et al (Bao et al., 2019) and Yang et al (Yang et al., 2020) in the context of video person re-ID. Recent advancements (Wang et al., 2020a; Zhang et al., 2021b; Nguyen et al., 2021) have explored the application of GCNs on local semantics to leverage the strengths of local information and topology modeling. These methods incorporate pose estimation to form natural graphs of key points or construct graphs of human parts through clustering or partitioning. Subsequently, they extract higher-level semantics to enhance local features or improve retrieval performance. In summary, previous methods have either relied on coarse-grained poses and stripes or attempted to enhance fine-grained human parts. However, enhancing part features alone does not fully address the underlying issues, as the challenge often lies in intra-instance gaps rather than insufficient discriminative power of part features. A GCN-based method utilizing coarse-grained human parts is briefly illustrated in Fig. 1(b).

In this paper, we introduce the Instance-level Consistent Graph (ICG) framework, which leverages the strengths of both fine-grained clustered human parts and graph modeling to explore robust structural features. The overall diagram of the ICG framework is illustrated in Fig. 2. ICG addresses the challenges of part-based person re-ID by focusing on two key aspects: extracting aligned part features and constructing a descriptor robust to misalignment.

For human parts extraction, we first employ spatial attention to separate the foreground from the background, effectively suppressing non-instance noise. By applying a threshold to the normalized feature map after attention, we obtain the foreground mask. Subsequently, we design an unsuper-

vised clustering method to localize pixel-wise human parts, including personal belongings. The K-means algorithm assigns pseudo-labels to each pixel within the foreground, clustering samples of the same person to generate human parts masks. Following this, foreground features and part features are obtained by masking the global feature extracted using a common backbone.

Building upon the clustered human parts, we propose a flexible structure graph where each node represents the features of one part. We then utilize GCNs to construct a descriptor for the structure of human parts. The graph allows insufficient or misaligned part features to interact with other nodes, enabling the flexible combination of features and consistent representation even with incomplete part features. In essence, ICG establishes a tolerant graph with clustered part features, relaxing the consistency assumption of re-ID from the part-level to the instance-level. The contributions of this work are summarized as follows:

- We introduce an attention-based foreground mask and pixel-wise human parts clustering to reduce misalignment. This approach suppresses non-instance noise and facilitates the extraction of fine-grained part features without requiring additional supervision.

- We propose the Instance-level Consistent Graph (ICG), which captures instance-level consistent structural information to tolerate part feature incompleteness. This is achieved by constructing a flexible structure graph with nodes representing part features.

- Extensive experiments validate the effectiveness of ICG, demonstrating superior performance compared to most state-of-the-art methods on three widely used person re-ID datasets, and exhibiting remarkable improvements over the baseline.

## 2 METHODOLOGY

In this section, we provide a detailed overview of each module in the ICG framework. We begin by introducing an attention-based foreground mask that effectively separates the foreground from non-instance noise. Following this, we discuss the pixel-wise human parts clustering process, which is conducted within the foreground mask to extract fine-grained part features. Additionally, we describe the construction of a flexible structure graph using part features, enabling the learning of instance-level consistent structural information. Finally, we outline the loss function applied based on the proposed modules. The overall framework is depicted in Fig. 2.

### 2.1 ATTENTION-BASED FOREGROUND MASK

To extract the foreground and background masks of the image without relying on additional segmentation models, we utilize the attention-based foreground mask (AFM) comprising a spatial attention layer and $l_1$ normalization. This design is based on the observation that the foreground response of the feature map tends to be larger than the background response. The spatial attention layer enhances the distinguishability between background and foreground responses by increasing the attention value of pixels. Subsequently, the foreground mask is derived by applying an intermediate threshold for binary classification based on the $l_1$ normalization of pixel channel features. The procedural steps are illustrated in Fig. 3.

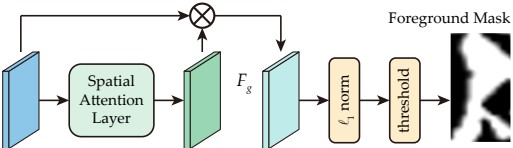

Figure 3: Diagram of the attention-based foreground mask.

Our ICG utilizes BoT (Luo et al., 2019) as the baseline. For an image $x_i$, the backbone network's mapping function (denoted as $f_\theta$) outputs the global feature mapping:

$$F_{\theta,g}^{c \times h \times w} = f_\theta(x_i) \tag{1}$$

where $\theta$ represents the backbone network's parameters, while $w$, $h$, and $c$ denote the width, height, and number of channels, respectively. The global features are then multiplied with the spatial attention layer parameters to generate size-invariant augmented features:

$$F_g^{c \times h \times w} = F_{\theta,g}^{c \times h \times w} \cdot S_A^{h \times w} \tag{2}$$

where $S_A^{h \times w}$ is learned by training, and the structure of the spatial attention layer is drawn as Fig. 4.

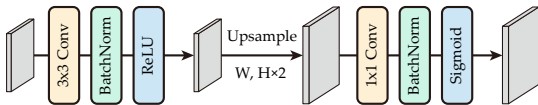

Figure 4: The spatial attention layer.

In the AFM, the pixels of all $F_g$ belonging to the same person (i.e., instance) are grouped into foreground or background based on their activation, following the principle that foreground pixels exhibit a higher response than background pixels (Ma et al., 2019). For a channel feature $F_g(x, y)$ at spatial location $(x, y)$, represented as a $c$-dimensional vector, the $l_2$ norm of $F_g(x, y)$ serves as the activation value for the pixel $(x, y)$. The activations of all pixels in $F_g$ are normalized by dividing the maximum activation value:

$$F_{l_1}(x, y) = \frac{\|F_g(x, y)\|_2}{\max_{(i,j)} \|F_g(i, j)\|_2} \tag{3}$$

where $(i, j)$ represents any position in $F_g$, and the maximum value of $F_1(x, y)$ is normalized to 1. During experimentation, a threshold of 0.5 is applied, categorizing pixels above this threshold as foreground and others as background.

Subsequently, the network learns the foreground confidence map $P_f^{h \times w}$ based on the generated mask. $P_f^{h \times w}$ serves as the true foreground label, enabling the network to prioritize the foreground region in the image. This process yields the foreground feature map $F_f^{c \times h \times w}$, as depicted in Eq. 4.

$$F_f^{c \times h \times w} = F_g^{c \times h \times w} \cdot P_f^{h \times w} \tag{4}$$

## 2.2 PIXEL-WISE HUMAN PARTS CLUSTERING

The key concept for achieving adaptive extraction of human parts at the pixel level involves determining the part to which each pixel in the foreground feature map $(x, y)$ belongs and generating the probability value of the pixel belonging to each part. This process is illustrated in the pixel-wise human parts clustering (PPC) diagram shown in Fig. 5. The probability values of all pixels form $K$ confidence maps $P_1^{h \times w}, \ldots, P_K^{h \times w}$, representing the network's learning objective. We denote $P_k(x, y)$ as the confidence that a pixel at $(x, y)$ belongs to semantic part $k$, where $k \in 1, \ldots, K$. It is important to note that personal objects are also treated as parts of the person. Ideally, if part $k$ is obscured in the image, $P_k(x, y) = 0$ should hold for all $(x, y)$, ensuring that the network does not generate representations of invisible parts. Ultimately, each body part comprises a collection of confidence-weighted pixel-wise representations. The PPC module takes the features of

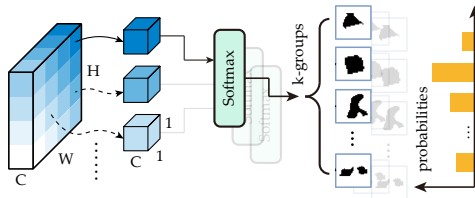

Figure 5: Diagram of pixel-wise human parts clustering.

all foreground regions belonging to an instance, specifically, individuals with the same ID, as input.

Subsequently, these features are split along the spatial dimension to create multiple one-dimensional features of varying sizes. These features serve as input for the K-means clustering after normalization as follow:

$$F_{l_2}(x, y) = \frac{F_g(x, y)}{\|F_g(x, y)\|_2} \tag{5}$$

K-means clustering then performs multi-class classification to assign semantic labels. Labels are assigned in a top-to-bottom order as follows: 0 represents the background pseudo-label, while $1, 2, ..., K$ denote the head, upper body, non-human part, lower body, and shoes, respectively.

After obtaining pseudo-labels for pixels in these local semantic regions, the network proceeds to map out the part features based on global features. Initially, the number of channels in the foreground features is reduced to $K + 1$ dimensions through convolutions, aligning with the total number of clusters. Subsequently, the pixel values of the $K + 1$ channels are subjected to softmax classification along the channel dimension, resulting in the generation of $K + 1$ confidence maps. Each probability map corresponds to a set of pixels in the same local semantic region. Denoting the convolutions as $f_{conv}$, the intermediate features are obtained as Eq. 6:

$$F_c^{k \times h \times w} = f_{conv}\left(F_{l_2}^{c \times h \times w}\right) \tag{6}$$

The feature is then processed using softmax operation and categorized into $K + 1$ classes, where $k = 0$ represents the background, resulting in the semantic probability graph for each part as shown in Equation 7.

$$P_k^{h \times w}(x, y) = \text{softmax}(F_{l_2}^{k \times h \times w}(x, y))$$
$$= \frac{\exp\left(W_k^T F_{l_2}^{k \times h \times w}(x, y)\right)}{\sum\limits_{i=0}^{K-1} \exp\left(W_i^T F_{l_2}^{k \times h \times w}(x, y)\right)}, k = 0, 1, ...K, (x, y) \in (w, h) \tag{7}$$

These $K + 1$ confidence maps are utilized to derive the feature maps for each part $F_k^{c \times h \times w} = F_g^{c \times h \times w} \cdot P_k^{h \times w}$, where $k \in 1, \ldots, K$. Subsequently, the foreground probability map is obtained by aligning pixels and summing them excluding the background:

$$P_f^{h \times w} = \sum_{k=1}^{K-1} P_k^{h \times w}, k = 1, \ldots, K \tag{8}$$

This foreground feature is then obtained by multiplying it with the global feature $F_g$:

$$F_f = F_g \cdot P_f^{h \times w} \tag{9}$$

Thus, three features of the pedestrian are obtained: the global features $F_g$ with foreground emphasis, the foreground features $F_f$, and the part features $F_{part-k}$.

## 2.3 FLEXIBLE STRUCTURE GRAPH

To extract structural information from the correlation between part features, we introduce the flexible structure graph (FSG) illustrated in Figure 6. Initially, the part features obtained from the PPC module serve as input. Subsequently, after initializing the correlation matrix $M$ and the GCN containing multiple hidden layers, the features are iteratively propagated between nodes. The final output is a one-dimensional feature vector $F_s$ of size $1 \times 2048$, which represents the structural information of the person.

The FSG is represented as $G = (V, E)$, where $V = (v_1, v_2, \ldots, v_N)$ comprises $N$ nodes corresponding to human parts. Each node represents one part of a person and is initialized using a feature vector $D$-dims $x_v$. The graph includes an adjacency matrix denoted as $M \in E^{N \times N}$. The correlation matrix $M$ takes the following form:

$$M = \begin{bmatrix} v_{1,1} & v_{1,2} & \ldots & v_{1,N} \\ v_{2,1} & v_{2,2} & \ldots & v_{2,N} \\ \vdots & \vdots & \ddots & \vdots \\ v_{N,1} & v_{N,2} & \ldots & v_{N,N} \end{bmatrix} \tag{10}$$

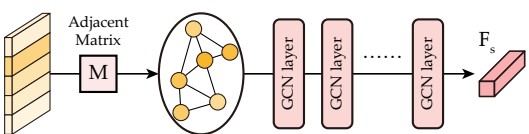

Figure 6: Diagram of the flexible structure graph.

We assume that parts are reliably identified among query images in the training set. Thus, for all elements of the matrix $M$, the value $v_{i,j}$ can be interpreted as follows: if the body part feature $F_i^q$ is identified, then the connection weight between $F_j^q$ and $F_i^q$ is $v_{i,j}$, with a maximum value of 1.

The GCN defines a multilayer propagation process on the graph $G$. Specifically, each layer in the GCN is represented as a function $f(X, M)$, which updates node representations by propagating information between them. Here, $X \in \mathbb{R}^{N \times D_w}$ represents the input node matrix, where each row corresponds to a node. $H^{(k)}$ denotes the feature matrix after passing the input node $X$ through the $k$-th GCN layer. We adopt the GCN formulation proposed by Kipf and Welling (Kipf & Welling, 2017), which takes node features $H^{(k)} \in \mathbb{R}^{N \times d}$ and the correlation matrix $M$ as input, transforming them into $H^{(k+1)} \in \mathbb{R}^{N \times d'}$ through the GCN layer. Each GCN layer can be represented as:

$$H^{(k+1)} = \text{LeakyReLU}(\hat{M} H^{(k)} \theta^{(k)}) \tag{11}$$

where $\theta^{(k)} \in \mathbb{R}^{d \times d'}$ is the trainable weight matrix of the $k$-th layer, and $\hat{M}$ is the normalized correlation matrix, which can be formulated as:

$$\hat{M} = (I + D)^{-\frac{1}{2}} (M + I)(I + D)^{-\frac{1}{2}} \tag{12}$$

$D$ represents the diagonal matrix of $M$, augmented with the unit matrix $I \in \mathbb{R}^{N \times N}$ to enforce self-loops in $G$. The objective of the GCN for feature extraction is to learn a set of parameters $\theta = \{\theta^{(1)}, \theta^{(2)}, \dots, \theta^{(k)}\}$, which map $X$ to the one-dimensional structural features $F_s$, capturing the distinctive structural information of a person for re-ID.

## 3 EXPERIMENTS

### 3.1 DATASET AND IMPLEMENTATION DETAILS

To demonstrate the effectiveness and generality of the proposed method, the network validates the performance on four widely used person re-identification datasets, Market-1501(Zheng et al., 2015), DukeMTMC-reID(Zheng et al., 2017), CUHK03-NPZhong et al. (2017) and MSMT17(Wei et al., 2018) datasets. Following the common practice of re-ID, methods use the Rank-1 cumulative matching characteristic (CMC) and the mean average precision (mAP) to evaluate performance, The mAP metric measures the combined accuracy of any ranked position.

Experiments are implemented on PyTorch 1.2.0, using a single NVIDIA Tesla V100 to accelerate training. First, the input images are resized to $128 \times 256 \times 3$ in data preprocessing, and the global feature map $F_g$ is 1/4 of the input size with 256 channels. Random cropping and random erasing (with a probability of 0.5) are applied as data augmentations to both the baseline method BoT(Luo et al., 2019) and our method ICG. Second, the ResNet50 backbone network is used to extract features, initialized from a pre-trained model on ImageNet. The Warmup policy is adopted for the first 10 epochs, in which the learning rate increases from $3.5 \times 10^{-5}$ to $3.5 \times 10^{-4}$ and then decreases back. The learning rate is reduced by a factor of 0.1 during epochs $40-70$. The model converges after 120 epochs. The batch size is set to 64, and the training is carried out using the Adam optimizer.

When clustering the feature maps to generate pseudo-labels, we used k-means(Kummamuru & Murty, 1999) as the clustering algorithm and re-clustered every $n$ epochs, which is a trade-off between parameter updating and pseudo-label generation. For the actual training, $n = 1$ is simply set. Instance-level clustering is applied in training only, and the inference time does not increase.

### 3.2 COMPARISON WITH STATE-OF-THE-ART METHODS

In this paper, the proposed method was compared with the state-of-the-art methods and the results are shown in Table 1, including methods utilizing local features alignment, attention mechanism,

spatial-temporal multi-scale fusion, semantic model-driven features and transformer structure. The MSMT17 dataset contains a large number of incomplete person images and many occlusion scenes, on which comparative experiments are conducted with other algorithms to demonstrate the performance of the proposed algorithms in complex scenarios. In table 1, the image input size is 128 $\times$ 256 $\times$ 3 for all methods, and the experimental results use the original data from the references. The results of the BoT algorithm on MSMT17 were obtained through training, and the training environment for BoT is the same as the conditions of ICG.

Table 1: Performance comparison results with state-of-the-art algorithms

| Method | Market-1501 | | DukeMTMC-reID | | MSMT17 | | CUHK03 | |
|---|---|---|---|---|---|---|---|---|
| | Rank-1 | mAP | Rank-1 | mAP | Rank-1 | mAP | Rank-1 | mAP |
| PCB+RPP(Sun et al., 2018) | 93.8 | 81.6 | 83.3 | 69.2 | 68.2 | 40.4 | 61.3 | 54.2 |
| CDPM(Wang et al., 2020b) | 95.2 | 86.0 | 88.2 | 77.5 | - | - | 75.8 | 71.1 |
| AOPS(Jin et al., 2022) | 94.6 | 85.3 | 87.5 | 76.3 | - | - | - | - |
| BPBReID(Somers et al., 2023) | 95.1 | 87.0 | 89.6 | 78.3 | - | - | - | - |
| HACNN(Li et al., 2018) | 91.2 | 75.7 | 80.5 | 63.8 | - | - | 44.4 | 41.0 |
| reID-NAS(Zhou et al., 2022) | 95.1 | 85.7 | 88.1 | 74.6 | 79.5 | 53.3 | - | - |
| AGW(Ye et al., 2022) | 95.1 | 87.8 | 89.0 | 79.6 | - | - | 63.6 | 62.0 |
| MHSA-Net(Tan et al., 2023) | 94.6 | 84.0 | 87.3 | 73.1 | - | - | 75.6 | 72.7 |
| MSINET(Gu et al., 2023) | 95.3 | 89.6 | - | - | 80.7 | **59.5** | - | - |
| ISP(Zhu et al., 2020) | 95.3 | 88.6 | 89.6 | 80.0 | - | - | 76.5 | 74.1 |
| GPS(Nguyen et al., 2021) | 95.2 | 87.8 | 88.2 | 78.7 | - | - | - | - |
| PGFA$_v$2(Miao et al., 2022) | 92.7 | 81.3 | 86.2 | 72.6 | - | - | - | - |
| OSNet(Zhou et al., 2019) | 94.8 | 84.9 | 88.6 | 73.5 | 78.7 | 52.9 | | |
| GASM(He & Liu, 2020) | 95.3 | 84.7 | 88.3 | 74.4 | 79.5 | 52.5 | - | - |
| PFE(Zhong et al., 2021) | 95.1 | 86.2 | 88.2 | 75.9 | 79.1 | 52.3 | 71.6 | 68.6 |
| OCLSM(Wang et al., 2021) | 94.6 | 87.4 | 87.7 | 79.0 | 78.8 | 57.0 | 71.0 | 68.3 |
| FA-Net(Liu et al., 2021) | 95.0 | 84.6 | 88.7 | 77.0 | 76.8 | 51.0 | - | - |
| PAT(Li et al., 2021) | **95.4** | 88.0 | 88.8 | 78.2 | - | - | - | - |
| TransReID(He et al., 2021) | 95.0 | **88.9** | 90.6 | 82.2 | - | - | - | - |
| AAformer(Zhu et al., 2023) | 95.4 | 88.0 | 90.1 | 80.9 | - | - | **77.6** | **74.8** |
| DAAT(Lu et al., 2023) | 95.1 | 88.8 | 90.6 | 82.0 | - | - | - | - |
| BoT-baseline(Luo et al., 2019) | 94.5 | 85.9 | 86.4 | 76.4 | 79.8 | 56.2 | 73.6 | 70.8 |
| **ICG (Ours)** | **95.4** | **88.9** | **91.4** | **82.4** | **81.6** | **59.5** | 76.9 | 74.4 |

According to Table 1, this method achieves optimal results for the mAP/Rank-1 metric on the Market-1501, DukeMTMC-reID and MSMT17 datasets. It is worth noting that, compared with the baseline model BoT(Luo et al., 2019), the Rank-1 and mAP metric improved by 0.9%/3.0%, 5.0%/6.0%, 1.8%/3.3%, respectively. This validates that the extension of the proposed method to the baseline model is effective.

This proposed method outperforms the Transformer-based PAT(Li et al., 2021) method. PAT infers the occlusion region based on encoder-decoder structure, and the results is not as intuitive as the proposed method using ID information and PPC to extract semantic region. Furthermore, the method requires a huge amount of computation, high cost in training and reasoning time, and the model is less lightweight than the ResNet-50 model used in our approach.

## 3.3 ABLATION STUDY

In this section, the effects of the AFM, PPC, and FSG on the proposed method are investigated.

Table 2: Ablation experiment results (K = 6)

| Baseline | AFM | PPC | FSG | Market-1501 | | DukeMTMC-reID | | MSMT17 | |
|---|---|---|---|---|---|---|---|---|---|
| | | | | Rank-1 | mAP | Rank-1 | mAP | Rank-1 | mAP |
| √ | | | | 94.5 | 85.9 | 86.4 | 76.4 | 79.8 | 56.2 |
| √ | √ | | | 94.8 | 86.4 | 87.3 | 77.7 | 80.2 | 57.4 |
| √ | √ | √ | | 94.7 | 87.2 | 90.0 | 80.1 | 80.4 | 58.6 |
| √ | √ | √ | √ | **95.4** | **88.9** | **91.4** | **82.4** | **81.6** | **59.5** |

**Attention-based Foreground Mask.** Due to the influence of backgrounds on pedestrian images, which hinders the network's effective utilization of valuable foreground features, the proposed algorithm introduces a foreground enhancement module to amplify and extract the foreground regions. As shown in the second row of Table 2, the introduction of this module led to improvements of 0.5%/0.3%, 0.7%/0.9%, and 0.5%/0.8% in mAP/Rank-1 performance across various datasets. This validates the efficacy of the AFM. Furthermore, during testing, a visual representation can be obtained by comparing the feature response maps of the backbone network with those processed by the AFM, as illustrated in Figure 7 below. The normalized feature maps reveal that, post-enhancement by the AFM, the feature map responses tend to concentrate more on foreground regions, exhibiting a notable reduction in background responses.

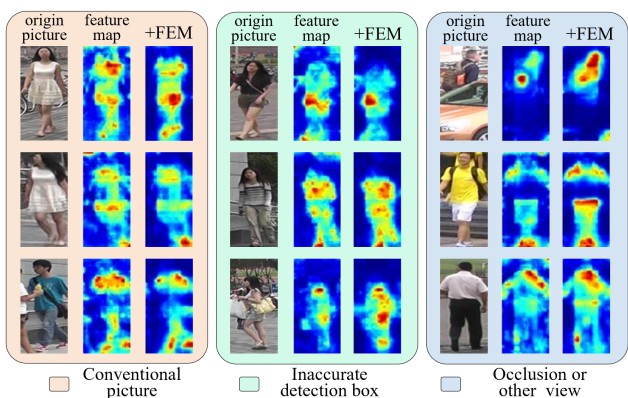

Figure 7: Foreground feature maps acquired from AFM in various scenarios

**Pixel-wise Human Parts Clustering.** Table 2 illustrates that the performance gains in mAP/Rank-1 for each dataset were 1.3%/0.2%, 3.7%/3.6% and 2.0%/1.4%, respectively, with the addition of PPC compared to the Baseline baseline model. The performance gains are significantly higher than those achieved by the foreground enhancement. This disparity can be attributed to the ability of PPC to provide more fine-grained local features, which are evidently more effective than a single global feature.

Intuitively, the number of cluster centers, denoted as $K$, determines the granularity of aligned parts when generating part pseudo-labels. In this approach, multiple part regions are obtained by clustering from top to bottom. The larger the value of $K$, the smaller the pixel share of each region, resulting in finer granularity. Consequently, the PPC generates different numbers of confidence maps for the classification of pixel channel features. In order to explore the influence of the number of clustering centers K on the network performance, the ablation experiment is conducted as Table 3.

Table 3: Influence of the number of clustering centers $K$

| $K$ | Market-1501 | | DukeMTMC-reID | | MSMT17 | | CUHK03 | |
|---|---|---|---|---|---|---|---|---|
| | Rank-1 | mAP | Rank-1 | mAP | Rank-1 | mAP | Rank-1 | mAP |
| 3 | 87.9 | 94.9 | 81 | 90.2 | 58.1 | 81.1 | 72.6 | 74.9 |
| 4 | 87.4 | 94.8 | 81.7 | 90.4 | 58.8 | **81.7** | 72.3 | 74.8 |
| 5 | 87.8 | 94.7 | 81.9 | 91.2 | 58.2 | 80.9 | 72 | 75 |
| 6 | **88.9** | **95.4** | **82.4** | **91.4** | **59.5** | 81.6 | **74.4** | **76.9** |
| 7 | 88.5 | 95.3 | 81.8 | 90.8 | 58.5 | 80.9 | 73.6 | 76.6 |

From Table 3, the model achieved near-optimal performance at K=6. To approximate real-life scenarios, images often include personal belongings such as backpacks. When the number of clusters is set to K=4, the generated local semantic regions may be relatively accurate, leading to a local optimum. However, when the number of clusters increases to K=7, the granularity of the generated regions becomes too fine, resulting in less effective local features for pedestrians and ultimately degrading network performance. At K=6, personal belongings are identified with the highest probability.

**Flexible Structure Graph.** The FSG can extract local feature relations and obtain unique structural features of a person. To verify the effectiveness of this module, Table 2 shows that the mAP/Rank-1 performance gains are 1.9%/0.6%, 4.1%/3.7% and 2.6%/2.3% when FSG is added compared to the baseline model. This demonstrates the effectiveness in FSG for modeling correlations between local features and that structural information is also representative of the intrinsic feature representation of a person.

The Figure 8 illustrates the visualization of the adjacency matrix for part features in FSG when K=6, and a higher value in the adjacency matrix indicates a stronger relationship between features. The local semantic regions of the two pedestrians in Figure 8(a) and Figure 8(b) are very close, indicating that their respective local features are similar. However, the adjacency matrices generated by FSG are different, demonstrating that the contextual features of the two similar pedestrians are distinct. As observed in Figure 8(d), the adjacency matrix connection values of potential personal belongings region and surrounding connected regions have a large value, indicating a strong relationship between personal belongings and the body region.

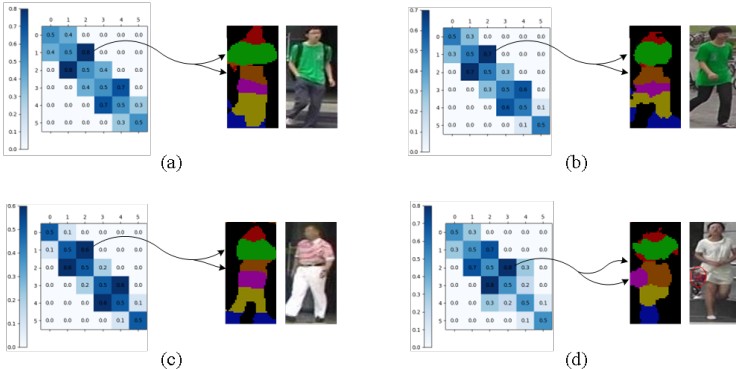

Figure 8: The visualization of adjacency matrix in FSG

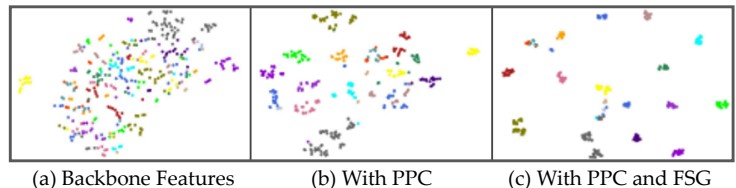

(a) Backbone Features      (b) With PPC      (c) With PPC and FSG

Figure 9: t-SNE visualization results

The person features are visualized to highlight the mutually reinforcing effect of the semantic unit adaptive module and the graph convolution module of this method, as shown in Figure 9. It can be observed that the features, which initially may not be very classifiable, are well-clustered after being enhanced by the two modules, demonstrating the effectiveness of the proposed method in classification.

With all modules added, the proposed method achieves optimal results with mAP/Rank-1 performance gains of 3.0%/0.9%, 6.0%/6.0% and 3.6%/3.3%. It can be verified that after the considerable performance achieved by the local feature approach, the performance is further improved by FSG, i.e., the correlation modeling between local features provides richer feature information and is the main factor that the conventional approach cannot break through the performance when facing part misalignment. PPC and FSG complement each other, expressing richer correlation features between human parts and enabling the network to achieve better performance.

### 3.4 MODEL COMPLEXITY ANALYSIS

The Table 4 presents a comparison of the computational complexity of the proposed ICG framework with other leading algorithms in terms of model size, floating-point operations, and performance on

Market-1501 Dataset. While TransReID achieves performance comparable to our proposed algorithm, its model is more complex. Similarly, MFA incorporates motion information at the feature map level, resulting in higher model complexity and computational cost. In contrast, the ICG framework introduces three simple yet effective modules, achieving superior performance with reduced complexity.

Table 4: Analysis of the model complexity and performance on Market-1501 dataset

| Algorithm | venue | mAP | Rank-1 | Parameters(M) | FLOPs(G) |
|---|---|---|---|---|---|
| OSNet(Zhou et al., 2019) | ICCV19 | 84.9 | 94.8 | 2.2 | 0.98 |
| Auto-ReID(Quan et al., 2019) | ICCV19 | 85.1 | 94.5 | 13.1 | 2.05 |
| MFA(Gu et al., 2022) | TIP22 | - | - | 84.0 | 20.06 |
| TransReID(He et al., 2021) | ICCV21 | **88.9** | 95.0 | - | 22.58 |
| TR-AMG-Base-Head25(Mao et al., 2023) | TMM23 | 88.5 | 95.0 | 21.3 | 16.2 |
| ICG(Ours) | - | **88.9** | **95.4** | 18.9 | 7.3 |

## 3.5 RETRIEVAL RESULTS

Figure 10 shows the retrieval results of the baseline (Luo et al., 2019) and the proposed method Top-10, which shows that the proposed method effectively addresses part misalignment in person re-identification. Furthermore, based on the results in group 4, we can observe the advantages of the FSG in extracting structural features. In the case of pedestrian clothes changing, the structural features between the person's part features and their personal belongings can provide more effective cues.

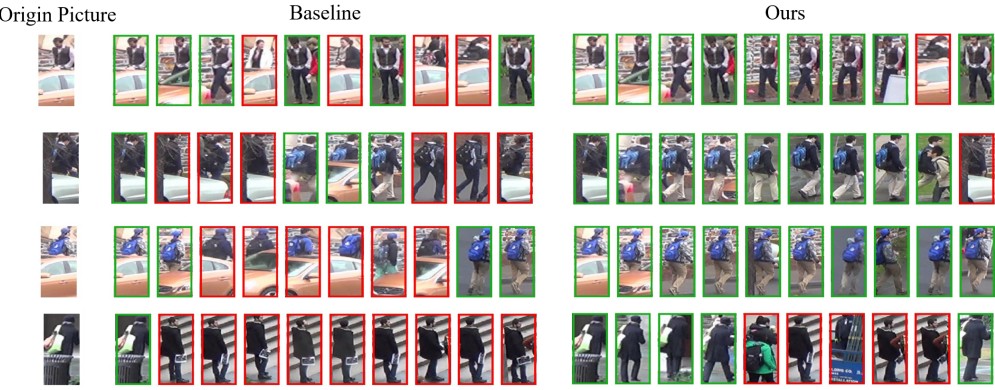

Figure 10: Retrieval results of the baseline and our method

## 4 CONCLUSION

To address the challenge of part misalignment caused by feature incompleteness in person re-ID, this paper first reviewed existing methodologies focusing on distinct human parts and introduced the concept of graph modeling. Recognizing the benefits of both approaches, we proposed the Instance-level Consistent Graph (ICG) framework, which integrates several key components. Initially, an attention-based foreground mask is applied to suppress background noise, followed by pixel-level human parts clustering within the foreground. This process enables the precise segmentation of fine-grained human parts. Subsequently, the constructed flexible structure graph employs nodes representing part features, facilitating interaction between them. This graph-based approach ensures the extraction of a robust structural feature that is insensitive to variations in node arrangements or absence. Through extensive experiments conducted on three widely used person re-ID datasets, we demonstrated the superiority of ICG over both baseline methods and state-of-the-art approaches. The results highlight the effectiveness of ICG in relaxing the matching condition from part-level to instance-level while maintaining distinctive part features, thereby advancing robust person re-ID methods.

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

## A APPENDIX

### A.1 METHODOLOGY

#### A.1.1 SIMILARITY MEASURE

During the testing phase, a total of four types of features can be obtained for each image to be queried: global feature $F_{global}$, foreground feature $F_{foreground}$, semantic feature $F_{part}$, and graph features $F_{graph}$. The proposed network in this paper employs cosine distance between features to measure the similarity between the query image and the image to be queried.

Since the number of part-level semantic features is adaptive and some part-level semantic areas are invisible when occluded, only visible part-level semantic features are selected for distance calculation. The total feature distance is divided into two parts: fixed global features, foreground features and graph features, and variable number of part-level semantic features.

Let $D(\cdot, \cdot)$ denote the cosine distance between features. The distances for global features, foreground features, graph features, and the $k$th semantic features of the query image $q$ and the image $g$ are:

$$\begin{cases} d_{global} = D(F_{global}^q, F_{global}^g) \\ d_{fore} = D(F_{fore}^q, F_{fore}^g) \\ d_{graph} = D(F_{graph}^q, F_{graph}^g) \\ d_{part\text{-}k} = D(F_{part\text{-}k}^q, F_{part\text{-}k}^g) \end{cases} \tag{13}$$

Here, the superscripts $q$ and $g$ represent the query image and the image to be queried respectively. Let $l_k$ denotes the visibility of the $k$th local semantic feature, and let $e_k$ denotes the existence of the distance $d_{part\text{-}k}$.

$$e_k = l_k^q \cdot l_k^g = \begin{cases} 1, \text{if } F_{part\text{-}k}^q \text{ and } F_{part\text{-}k}^g \text{ both exist} \\ 0, \text{otherwise} \end{cases} \tag{14}$$

The final average similarity distance is calculated as follows:

$$d = \frac{d_{global} + d_{fore} + d_{graph} + \sum\limits_{k=1}^{K-1} e_k d_{part\text{-}k}}{\sum\limits_{k=1}^{K-1} e_k + 3} \tag{15}$$

#### A.1.2 LOSS FUNCTION

In the training phase, the loss function of the network consists of three components: loss of AFM, PPC, and composed features.

The loss of AFM is the cross-entropy loss constituted by the foreground confidence map $P_f(x, y)$ and the foreground mask:

$$P_f(x, y) = \frac{\exp\left(W_f^T F_g(x, y)\right)}{\sum\limits_{i=0}^{1} \exp\left(W_i^T F_g(x, y)\right)} \tag{16}$$

$$L_{sep} = \sum\limits_{x,y} -\log P_{f_i}(x, y) \tag{17}$$

The number of channels was changed to $K$ dimensions based on the foreground feature map $M_g$, using a $1 \times 1$ convolution kernel as a linear layer, as described above, for each pixel site prediction. With the softmax classifier, $K$ confidence maps are obtained, which are expressed as:

$$P_k(x, y) = \frac{\exp\left(W_k^T F_f(x, y)\right)}{\sum\limits_{i=1}^{K} \exp\left(W_i^T F_f(x, y)\right)} \tag{18}$$

where $k \in \{0, \cdots, K-1\}$, $W$ are the parameters of the linear layer.

The $K$-dim vector composed of $P(x, y)$ at the spatial location $(x, y)$ is optimized with the pseudo-label $k_i$ obtained from clustering using cross-entropy loss to obtain the loss of PPC as:

$$L_{par} = \sum_{x,y} -\log P_{k_i}(x, y) \tag{19}$$

On the other hand, the features were processed through the BNNeck module following the optimization of BoT Luo et al. (2019). The composed feature $F_c$ for retrieval includes global features $F_g$, foreground features $F_f$, part features $F_p$, and structural features $F_s$. Each type of features corresponds to a loss group $L$ consists of the triplet loss, center loss, ID classification loss with label smoothing Luo et al. (2019). Let $\mathbb{L} = \{L_g, L_f, L_p, L_s\}$ is the set of losses for four types of features, the loss of the composed feature can be presented as:

$$L = L_{ID} + L_{tri} + L_{cen}, L \in \mathbb{L} \tag{20}$$

$$L_{feat} = L_g + L_f + L_p + L_s \tag{21}$$

The overall objective function is:

$$L_{opt} = \alpha_{feat} L_{feat} + \alpha_{sep} L_{sep} + \alpha_{par} L_{par} \tag{22}$$

Where each $\alpha$ is a balance weight, the experimental settings are 0.2, 0.1, and 0.1 for $\alpha_{feat}, \alpha_{sep}, \alpha_{par}$, respectively.

## A.2 ABLATION STUDY

### A.2.1 PIXEL-WISE HUMAN PARTS CLUSTERING

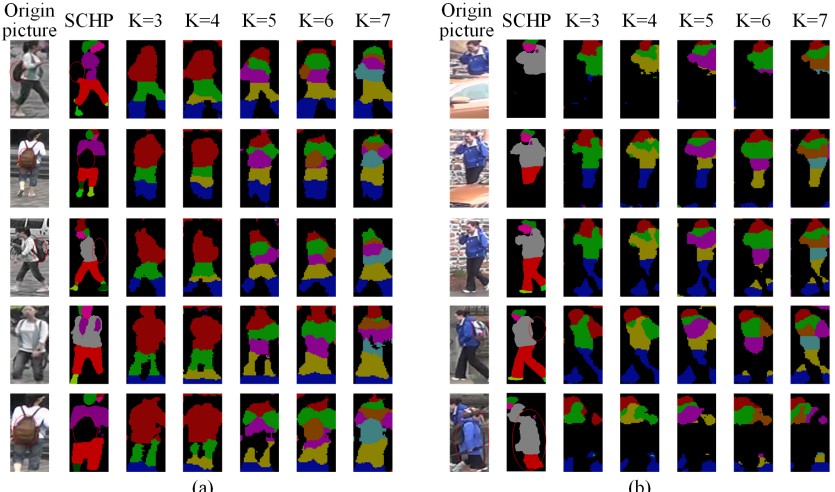

(a)                                        (b)

Figure 11: Human parts generated by PPC (red circles represent personal belongings)

The visualizations of the impact of the number of clusters K on the generation of local semantic regions are as in Figure 11.

As shown, at K=6, potential personal belongings are effectively identified and incorporated into the pedestrian representation. Figure Figure 11(b) illustrates the parsing results under occlusion conditions. The first two rows show that the PPC module achieves performance nearly comparable to the SCHP algorithm, effectively distinguishing occluded areas. The results in the last row highlight the advantages of our approach in handling occlusion caused by other pedestrians. SCHP struggles to distinguish such cases, treating features of other pedestrians as interference. In contrast, our method clusters features from all images of the same ID, allowing information sharing across instances. This enables our model to discard features of other pedestrians as background, demonstrating the benefits of our approach. Moreover, observing each column in the figure, regardless of the value of K, the PPC module consistently ensures local semantic region alignment across images.

ICG was trained for 120 epochs, with labels updated every two cycles. In order to explore the change process of pseudo labels of local semantic regions with training, we visualized the pseudo label results of clustering. When K is 6, the change process of pseudo labels of each local semantic region is as Figure 12.

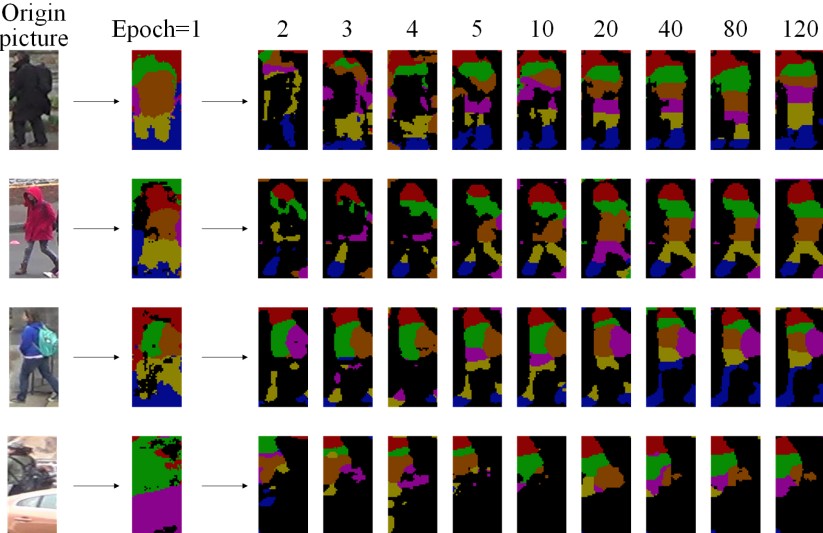

Figure 12: Visualization of PPC clustering with training epochs (red circles represent personal belongings)

As can be seen from Figure 12, as the training progresses, the clustering contour map of the pedestrian image shows a change process from chaos to clarity. As the training progresses, the global features are continuously optimized under the constraints of ID loss, focusing the response on the foreground area. At the same time, enhanced by the AFM module, the distinction between foreground and background areas is more obvious. Therefore, the subsequent local semantic regions gradually become correct, and the background is well excluded. In the end, the global features do not change much with the optimization process, and the local semantic regions also slowly remain unchanged, and local semantic regions with good precision are obtained. In addition, it can be noted that Figure 12(a) and Figure 12(b) show the gradient results when the pedestrian body is similar in large areas. It can be found that the semantic region gradually becomes complete, and it is close to the final effect image at least in 20 Epochs. Figure 12(c) shows the gradient result when the pedestrian includes the accompanying objects. It can be found that due to the large differences between the local areas of the pedestrian, the clustering effect is very fast. When Epoch is 1, the pedestrian's accompanying objects can be detected. Figure 12(d) shows that pseudo-labeling can also achieve good results in the case of occlusion. In short, the gradient process of the above pseudo-labels proves the effectiveness and feasibility of obtaining local semantic areas through feature map clustering.

## A.3 DEMONSTRATION OF EXPERIMENTAL RESULTS

### A.3.1 SEMATIC FEATURE MAPS VISUALIZATION

The following figure illustrates the semantic feature visualization results of our algorithm and compares them with pseudo-labels. The left-side results depict the local feature distribution of a typical pedestrian's various parts in the presence of background clutter. It can be observed that the pseudo-labels of the pedestrians are distributed from top to bottom, and the corresponding semantic feature response maps also follow this top-to-bottom arrangement. This validates the effectiveness of using pseudo-labels as semantic supervisory information. Only through this approach can the semantic features focus on local information. Upon comparing the semantic features with the corresponding areas of pseudo-labels, distinctions are evident. This divergence stems from the enforcement of ID classification loss on local semantic features, causing the predicted labels for semantic regions to outperform the pseudo-labels generated by clustering. When examining the comparison between pseudo-labels and foreground features, a strong correspondence is apparent. Additionally, the last two rows display how the foreground features of pedestrians eliminate the influence of the background, thereby enhancing recognition performance. On the right side, results are presented in scenarios containing pedestrians with potential personal items and occlusions. It can be observed that the localized semantic regions formed by our algorithm effectively exclude occluded regions while retaining the advantage of identifying potential personal items carried by pedestrians. This further reinforces the representational capacity of pedestrians and improves recognition effectiveness in complex scenarios.

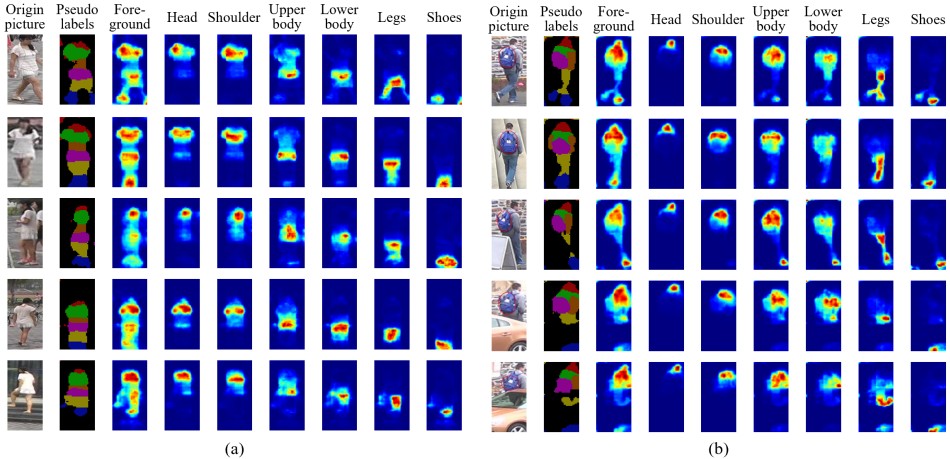

Figure 13: Pseudo-tags compared to the predicted part features response map.

### A.3.2 PRESENTATION OF LEARNED HUMAN PARTS

Since there is no pre-labeling of human parts for the re-ID dataset, this article uses the state-of-the-art parsing model SCHP Li et al. (2022) to parse the human parts in the training set to create four types of the ground-truth. Parts of the hat, hair, sunglasses, face, etc. are aggregated as the head, the left leg, right leg, socks, and trousers are aggregated as legs, the left shoe and right shoe are aggregated as shoes. Since the pre-trained model is unable to recognize personal belongings, that part is or evaluated. The accuracy of the pseudo-labeling on the training set and the semantic estimation on the test set are then evaluated using the region intersection over Union (IoU) ratio in semantic segmentation Li et al. (2022), as shown in Table 4.

From Table 5, we can find a conclusion that the accuracy of the semantic parts predicted by the trained network on the test set is mostly higher than that of the pseudo-partial labels on the training set for all datasets. This indicates that the generalization of our method on the test set is higher than that of the clustering method. The clustering method may produce bias in the clustering process, while the proposed network can ensure that the part features of multiple images are the same person as far as possible based on the loss of ID classification, so the final semantic part estimation of our

Table 5: Semantic parsing performance of PPC

| Evaluated Labels | Datasets | IoU(%) | | | |
|---|---|---|---|---|---|
| | | Head | Shoulder | Leg | Shoes |
| Learned (on train set) | Market | 65.4 | 54.7 | 67.2 | 55.2 |
| | Duke | 65.6 | 68.2 | 61.8 | 58.9 |
| | MSMT | 56.1 | 61.3 | 59.2 | 56.7 |
| Predicted (on test set) | Market | 66.6 | 55.2 | 67.9 | 56.2 |
| | Duke | 66.9 | 72.1 | 69.1 | 60.1 |
| | MSMT | 58.8 | 62.4 | 60.5 | 59.8 |

network is better than the pseudo-part labels of the clustering algorithm and more robust to part estimation.

To illustrate the learning process of clustering of semantic labels, Figure 14 shows how the pseudo-labels obtained from clustering become more refined with the training of the feature map at K=6. This demonstrates that the clustering of the parts of this method becomes better as the feature maps are generated iteratively. When K is the same, the results in the first row show that the colour of the same part remains the same for different images of the same person, which verifies the semantic consistency in this method.

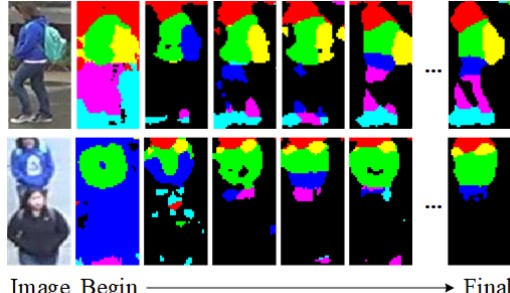

Image Begin ⟶ Final

Figure 14: The process of clustering when $K = 6$