# OpenReview forum: "Instance-level Consistent Graph With Unsupervised Human Parts for Person Re-identification"
_ICLR.cc/2025/Conference — ICLR 2025 Conference Withdrawn Submission_

### Official Review · Reviewer_jcgN · 2024-10-31

**Soundness:** 2
**Presentation:** 2
**Contribution:** 2
**Rating:** 6
**Confidence:** 5

**Summary:**

This paper introduces the Instance-level Consistent Graph (ICG) framework to improve person re-identification (re-ID) by addressing challenges such as misalignment, occlusion, and varying illumination. ICG employs an attention-based foreground mask to separate instances from non-instance noise, followed by pixel-wise clustering for extracting fine-grained human part representations. A graph convolutional network then organizes these part features into a flexible structure graph, enabling instance-level structural consistency and improving resilience to feature incompleteness. Extensive evaluations on three popular re-ID datasets demonstrate superior performance over state-of-the-art methods.

**Strengths:**

1.Effective Module Design: The integration of three core components—the attention-based foreground mask (AFM), pixel-wise human parts clustering (PPC), and flexible structure graph (FSG)—is systematically designed and demonstrates effectiveness through improved feature alignment and robustness.

2.Solid Empirical Validation: The paper provides a thorough evaluation, with experimental results showcasing clear improvements over baseline models across various datasets (e.g., Market-1501, DukeMTMC-reID, and MSMT17), demonstrating ICG’s ability to handle occlusions and alignment issues.

**Weaknesses:**

1.Engineering-focused: The method, though innovative, may appear incremental as it combines known techniques (attention, clustering, graph convolution) without fundamentally novel theoretical contributions. Further insights into ICG’s scalability or potential applications could strengthen the impact.

2.Limited Component Analysis: More detailed ablation studies on individual settings within each module (such as varying clustering levels within PPC or adjacency thresholds in FSG) could provide a clearer understanding of the specific impact of each component.

3.The paper’s overall approach is straightforward, but many modules consist of existing techniques without significant innovation, making this work largely incremental.

4.The method comparisons are somewhat outdated. In recent years (2023-2024), traditional re-ID methods have continued to make advancements, with many reaching mAP scores around 91-92 on Market-1501 without relying on pretrained weights like CLIP. The authors should include more recent benchmarks to better contextualize their results.

5.The method heavily depends on the quality of the masks, yet in Figure 8, visualizations reveal that many irrelevant areas (e.g., background) are still extracted alongside the person. This interference can disrupt alignment. To improve robustness, the authors should prioritize generating higher-quality masks that better isolate the target person.

6.The paper lacks an analysis of computational complexity, including trainable parameters and FLOPs. The authors should provide this analysis and offer comparisons to similar methods to give a clearer understanding of the model’s efficiency.

**Questions:**

see weakness

---

> ### Author Response · Authors · 2024-11-26
> **Official Response to Reviewer jcgN (PART I)**
>
> We sincerely appreciate your constructive and thoughtful feedback. Below are our supplementary experiments and responses:
>
> 1. Weakness 1
>
> Thank you for your valuable suggestion.
>
> While several modules in our framework rely on established methodologies, the proposed approach of leveraging a more flexible instance-level consistency for person re-identification, instead of pursuing strict part-level consistency, proves to be practical and effective.
>
> Our ICG framework utilized attention-based foreground mask to extract pedestrian masks, ensuring that subsequent operations are performed on enhanced foreground features. This approach effectively mitigates the issues of fine-grained human part misalignment and coarse-grained image block misalignment outlined in Figure 1 of our paper. Additionally, pixel-wise human parts clustering enables unsupervised clustering of pedestrian parts. In the PPC module, features from all images of the same ID are clustered together, ensuring that the model maintains instance-level consistency during the clustering process. Furthermore, the flexible structure graph constructs a dynamic graph for each query image. This graph uses the human parts and their labels obtained via K-means clustering as the initial graph matrix, which is then updated through multiple GCN layers. The GCN strengthens the relationships between correlated human parts and attenuates the effects of misalignment caused by partial occlusion, pose variations, and extreme illumination, thereby forming more robust pedestrian structural features.
>
> We have expanded the discussion in appendix to highlight potential applications of ICG in real-world scenarios (e.g., smart city surveillance, public transportation safety).  These use cases illustrate the practical impact and versatility of our approach.
>
> In addition, inspired by the reviewer fRrV,  based on our flexible instance-level consistency framework ICG, incorporating advanced clustering techniques, modern self-attention mechanisms, or graph learning approaches may further enhance the methodology.
>
> 2. Weakness 2
>
> We added an ablation experiment about the number of clustering centers $K$ in PPC module, and conducted further analysis on the impact of the number of clusters K on the generation of local semantic regions by visualization figure. The detailed explanation could be find in our response to fRrV (PART II) 3.Weakness 3
>
> 3. Weakness 3
>
> We sincerely thank you for pointing the potential limitations of relying on established methodologies. The motivation of our proposed ICG framework, as well as the realization ideas for its components, has been discussed in our response to weakness 1. The ICG framework introduces a more flexible instance-level consistency approach, and our experiments have demonstrated the feasibility and effectiveness of this framework.
>
> 4. Weakness 4
>
> Thank you for your insightful suggestion regarding the importance of including up-to-date results to prove the effectiveness of our method. We removed certain outdated methods from our comparison updated some recent methods. Specific modifications could be referred to response to Reviewer xMUt Weakness 2.
>
> Regarding the performance on the Market-1501 dataset, we would like to highlight two important observations:
>
> Performance Saturation: Many recent methods achieve mAP scores of 91-92 on Market-1501, indicating near-saturation. This limits the dataset’s ability to distinguish newer methods or reflect advancements on more complex datasets.
>
> Re-Ranking Impact: Many methods use re-ranking to boost mAP by 1-2 points, which can obscure the model’s true discriminative ability. To ensure fair comparison, we avoided re-ranking in our evaluation.

---

> > ### Author Response · Authors · 2024-11-26
> > **Official Response to Reviewer jcgN (PART II)**
> >
> > 5. Weakness 5
> >
> > AFM selectively enhances the foreground and suppresses the background at feature-level. The AFM module is based on the observation that the foreground response in feature maps tends to be larger than the background response. The spatial attention layer enhances the contrast between the foreground and background by increasing the attention values of foreground pixels. Through classification loss and part-parsing loss, the network is progressively guided to focus more on foreground features during learning.
> >
> > In the analysis on the impact of the number of clusters K on the generation of local semantic regions in appendix, we compared the effects of clustering algorithms with various K values against an additional semantic parsing model. SCHP algorithm is a popular semantic parsing model. Figure (b) illustrates the parsing results under occlusion conditions. The first two rows show that the PPC module achieves performance nearly comparable to the SCHP algorithm, effectively distinguishing occluded areas. The results in the last row highlight the advantages of our approach in handling occlusion caused by other pedestrians. SCHP struggles to distinguish such cases, treating features of other pedestrians as interference. In contrast, our method clusters features from all images of the same ID, allowing information sharing across instances. This enables our model to discard features of other pedestrians as background, demonstrating the benefits of our approach.
> >
> > 6. Weakness 6
> >
> > Thank you for your valuable suggestion. The computational complexity analysis could be referred to the response to Reviewer 7m9c (PART I) Weakness 1 third point.

---

> > > ### Comment · Reviewer_jcgN · 2024-11-29
> > > **Thanks for your reply, I will raise my score.**
> > >
> > > Thank you for your thoughtful response. Your reply has addressed most of my concerns; however, some points of understanding remain unresolved. Regarding Point 4, methods such as HAT, RGA-SC, PHA, and GLTrans outperform your approach on most datasets. However, comparing your method with frameworks that are either CNN-based or a hybrid of CNN and Transformer might be more appropriate. Therefore, I remain cautious about your response to Point 4.
> > >
> > > Additionally, in your reply to Weakness 3, I believe the novelty of your method is still insufficient. Many approaches already utilize attention mechanisms for feature selection, such as *Magic Tokens: Select Diverse Tokens for Multi-Modal Object ReID* (CVPR 2024). Considering the feedback from other reviewers, I am willing to adjust my score upward. However, due to the aforementioned issues, I maintain a degree of reservation.

---

> > > > ### Author Response · Authors · 2024-12-03
> > > > **Thanks for the further comments.**
> > > >
> > > > We sincerely appreciate your willingness to adjust score based on our improvements and responses. We acknowledge the lack of comparative analysis with hybrid CNN-Transformer architectures for person re-identification in our current work. This will be addressed and improved in our future revisions. Admittedly, our approach leans towards proposing a more flexible instance-level consistency framework, aiming at mitigating the challenges of precise part alignment faced by prior works. The modules in ICG are generalizable and easy to implement. In the future, we will refine our modules to better align with the specific characteristics of person re-identification tasks,  and present better performance. Thank you again for your support and valuable feedback!

---

### Official Review · Reviewer_7m9c · 2024-11-02

**Soundness:** 3
**Presentation:** 3
**Contribution:** 2
**Rating:** 5
**Confidence:** 4

**Summary:**

The paper introduces a newframework, Instance-level Consistent Graph (ICG), aimed at addressing the challenges of part misalignment and feature incompleteness in person re-identification tasks. The ICG framework innovatively integrates an attention-based foreground mask (AFM), pixel-wise human parts clustering (PPC), and a flexible structure graph (FSG) to extract robust structural features that are tolerant to variations in part arrangements or absences.

The AFM module enhances the foreground features by suppressing background noise, while the PPC module performs pixel-level clustering to segment fine-grained human parts within the foreground. The FSG then constructs a graph where each part feature is treated as a node, allowing for feature interaction and consistent representation even with incomplete parts. Extensive experiments on three major person re-ID datasets demonstrate that the ICG framework outperforms state-of-the-art methods and showcases significant improvements over the baseline model.

**Strengths:**

1. The concept of moving from strict part-level consistency to a more robust instance-level consistency is innovative and expands the possibilities for handling misalignment and occlusion in re-ID tasks.

2. The experiments are rigorous and well-designed, with performance metrics that are standard in the field. The paper demonstrates a significant improvement over the baseline and state-of-the-art methods, which speaks to the quality of the proposed approach.

3. The paper is well-written and organized, with a logical flow that makes it easy to follow. The introduction effectively sets the stage for the problem, the methodology is clearly described, and the results are presented in a manner that is easy to understand.

**Weaknesses:**

1. While the paper demonstrates strong performance on the three major datasets, it lacks a discussion on the generalizability of the ICG framework to other datasets or scenarios with different characteristics. Adding experiments on more diverse datasets could strengthen the paper's claims. The computational complexity of the ICG framework is not discussed.

2. The paper could improve by providing a more in-depth discussion on the limitations of the ICG framework. For instance, are there specific scenarios or types of occlusion where the method underperforms?

3. The paper could address potential ethical considerations and biases in the proposed system, especially since person re-identification has implications for privacy and surveillance. Discussing how the model handles different demographic groups and mitigating bias would be an important addition.

4. The typesetting of this paper seems unreasonable, and the content is not rich enough. For example, Figure 2 is placed at the bottom of the page, while Table 1 exceeds the width of the page. Also, this paper seems like it should further discuss additional experiments and ethical issues in an appendix, but it does not provide one.

5. Based on the current version of the paper, it seems that the paper is difficult to replicate due to a lack of sufficient detail.

In summary, while the paper makes some contributions to the field of person re-identification, there are areas where it could be improved. Addressing these weaknesses would not only strengthen the paper's claims but also provide a clearer path for future research and practical implementation.

**Questions:**

Will you make your code and model publicly available? This is important for the development of the field.

---

> ### Author Response · Authors · 2024-11-25
> **Official Response to Reviewer 7m9c (PART I)**
>
> We sincerely appreciate your constructive and thoughtful feedback. Below are our supplementary experiments and responses:
> 1. Weakness 1
> * Testing on additional datasets
>
> To address your concerns about generalizability, we conducted experiments on the CUHK03 dataset, another classic pedestrian re-identification dataset. We presented the performance comparison between the proposed ICG framework and state-of-the-art methods on MSMT17 dataset and CUHK03 dataset.  Most of the experimental results are derived from the literature and are summarized in the table below.
> | Algorithm      | MSMT17 (Rank-1) | MSMT17 (mAP) | CUHK03 (Rank-1) | CUHK03 (mAP) |
> |:--------------:|:---------------:|:------------:|:---------------:|:------------:|
> | PCB+RPP        |      68.2       |     40.4     |      61.3       |     54.2     |
> | CDPM           |        -        |       -      |      75.8       |     71.1     |
> | HACNN          |        -        |       -      |      44.4       |     41.0     |
> | reID-NAS       |      79.5       |     53.3     |        -        |       -      |
> | AGW            |        -        |       -      |      63.6       |     62.0     |
> | MHSA-Net       |        -        |       -      |      75.6       |     72.7     |
> | ISP            |        -        |       -      |      76.5       |     74.1     |
> | GASM           |      79.5       |     52.5     |        -        |       -      |
> | PFE            |      79.1       |     52.3     |      71.6       |     68.6     |
> | OCLSM          |      78.8       |     57.0     |      71.0       |     68.3     |
> | FA-Net         |      76.8       |     51.0     |        -        |       -      |
> | AAformer       |        -        |       -      |   **77.6**      |  **74.8**    |
> | BoT-baseline   |      79.8       |     56.2     |      73.6       |     70.8     |
> | ICG (Ours)     |   **81.6**      |  **59.5**    |      76.9       |     74.4     |
>
> In conclusion, the ICG framework achieves the best performance on the MSMT17 dataset and performs second only to AAformer on the CUHK03 dataset, demonstrating the effectiveness and Generalizability of the proposed approach.
> * Testing in real-world scenarios
>
> Due to time constraints, we were unable to collect and annotate a large-scale dataset from real-world scenarios. However, we tested the proposed ICG framework in practical surveillance scenarios. Specifically, we evaluated its performance under three indoor and outdoor cameras on the same pedestrian with varying viewpoints. The ICG framework successfully re-identified the same individual across all settings. While we cannot include test images in this response, we will provide the example in the appendix of the revised paper.
> * Model complexity analysis
>
> The follow table presents a comparison of the computational complexity of the proposed ICG framework with other leading algorithms in terms of model size, floating-point operations, and performance on Market-1501 Dataset. While TransReID achieves performance comparable to our proposed algorithm, its model is more complex. Similarly, MFA incorporates motion information at the feature map level, resulting in higher model complexity and computational cost. In contrast, the ICG framework introduces three simple yet effective modules, achieving superior performance with reduced complexity.
> | Algorithm                  |   mAP   | Rank-1 | Parameters (M) | FLOPs (G) |
> |:--------------------------:|:-------:|:------:|:--------------:|:---------:|
> | OSNet (ICCV19)              |  84.9   |  94.8  |      2.2       |    0.98   |
> | Auto-ReID (ICCV19)          |  85.1   |  94.5  |      13.1      |    2.05   |
> | MFA (TIP22)                |    -    |    -   |       84       |   20.06   |
> | TransReID (ICCV21)          | **88.9**|   95   |        -       |   22.58   |
> | TR-AMG-Base-Head25 (TMM23)|  88.5   |   95   |      21.3      |   16.2    |
> | ICG (Ours)                | **88.9**| **95.4**|     18.9      |    7.3    |
>
> 2. Weakness 2
>
> The appearance changes in pedestrians' clothing can affect re-identification performance. The ICG framework, based on instance consistency, segments human body parts using a clustering algorithm. If a pedestrian’s clothing changes, the part-level features may differ significantly, which could hinder matching.
>
> The accuracy of the foreground mask also plays a crucial role. Our attention-based mask, which avoids the need for an additional semantic parsing model, suffers from blurred edges, potentially including background information that affects performance.
>
> Additionally, pedestrian re-identification methods based on Transformers and infrared-visible fusion have advanced rapidly in recent years. While the ICG framework, relying on CNNs for part segmentation, shows strong results, we recognize the importance of exploring and integrating emerging techniques in future work.

---

> ### Author Response · Authors · 2024-11-25
> **Official Response to Reviewer 7m9c (PART II)**
>
> 5. Weakness 5
>
> Due to space limitations, the original manuscript lacked sufficient detail regarding the similarity measurement between query and test images, as well as the loss function. Here, we provide a brief explanation and will include more detailed descriptions in the appendix of the revised version, covering the output features of each module and the loss function setup. This will allow readers to better understand and replicate the proposed method.
>
> * Similarity measure
>
> During the testing phase, a total of four types of features can be obtained for each image to be queried: global feature ${F_{global}}$, foreground feature ${F_{foreground}}$, semantic feature ${F_{part}}$, and graph features $F_{graph}$. The proposed network in this paper employs cosine distance between features to measure the similarity between the query image and the image to be queried.
>
> Since the number of part-level semantic features is adaptive and some part-level semantic areas are invisible when occluded, only visible part-level semantic features are selected for distance calculation. The total feature distance is divided into two parts: fixed global features, foreground features and graph features, and variable number of part-level semantic features.
>
> Let $D(\cdot \text{ , }\cdot )$ denote the cosine distance between features, the distances for global features, foreground features, graph features, and the $k$th semantic features of the query image $q$ and the image $g$ are: ${d_{global}}=D(F_{global}^{q},F_{global}^{g})$, ${d_{foreground}}=D(F_{foreground}^{q},F_{foreground}^{g})$, ${d_{graph}}=D(F_{graph}^{q},F_{graph}^{g})$, ${d_{part-k}}=D( F_{part-k}^{q},F_{part-k}^{g})$.
>
> If the $k$th part-level semantic feature of the query image q and the query image g both exist, then $l_k^q⋅l_k^g=1$. In other cases, $l_k^q⋅l_k^g=0$. The final average similarity distance is:
>
> \begin{equation}
> 	d=(\sum\limits_{k=1}^{K-1}{e_{k}}{d_{part-k}}+( {d_{global}}+{d_{foreground}}+{d_{graph}}))/(\sum\limits_{k=1}^{K-1}{l_k^q⋅l_k^g}+3)
> \end{equation}
>
> * Loss function
>
> In the training phase, the loss function of the network consists of three components: loss of AFM, PPC, and composed features.
>
> The loss of AFM is the cross-entropy loss constituted by the foreground confidence map ${P_f}(x,y)$ and the foreground mask:
>
> \begin{equation}
> 	{L_{sep}}=\sum\limits_{x,y}{-}\log{P_{f_i}}(x,y)
> \end{equation}
>
> The number of channels was changed to $K$ dimensions based on the foreground feature map ${M_g}$, using a $1{\times}1$ convolution kernel as a linear layer, as described above, for each pixel site prediction. With the softmax classifier, $K$ confidence maps are obtained, which are expressed as ${{P}_{k}}(x,y)$
>
> The $K$-dim vector composed of $P(x,y)$ at the spatial location $(x,y)$ is optimized with the pseudo-label ${{k}_{i}}$ obtained from clustering using cross-entropy loss to obtain the loss of PPC as:
>
> \begin{equation}
> 	{{L}_{par}}=\sum\limits_{x,y}{-}\log {P_{k_i}}(x,y)
> \end{equation}
>
> On the other hand, the features were processed through the BNNeck module following the optimization of BoT. The composed feature $F_c$ for retrieval includes global features ${F_g}$, foreground features ${F_f}$, part features ${F_p}$, and structural features ${F_s}$. Each type of features corresponds to a loss group $L$ consists of the triplet loss, center loss, ID classification loss with label smoothing. Let $\mathbb{L} = { L_g, L_f, L_p, L_s }$ is the set of losses for four types of features, the loss of the composed feature can be presented as:
>
> \begin{equation}
>     L = L_{ID} + L_{tri} + L_{cen}, L \in \mathbb{L}
> \end{equation}
>
> \begin{equation}
> L_{feat}  = L_g + L_f + L_p + L_s % \sum_{L{\in}\mathbb{L}}{L}
> \end{equation}
> The overall objective function is:
>
> \begin{equation}
> 	L_{opt} = \alpha_{feat}L_{feat} + \alpha_{sep}L_{sep} + \alpha_{par} L_{par}
> \end{equation}
>
> Where each $\alpha$ is a balance weight, the experimental settings are 0.2, 0.1, and 0.1 for $\alpha_{feat}, \alpha_{sep}, \alpha_{par}$, respectively.

---

> > ### Author Response · Authors · 2024-11-25
> > **Official Response to Reviewer 7m9c (PART III)**
> >
> > 3. Weakness 3
> >
> > There are potential privacy risks associated with using demographic data in ReID tasks. (1) While current ReID research often uses implicit demographic information, such as images collected from different campuses in datasets like Market1501 and CUHK, it is possible to infer a pedestrian’s location and identity based on the background or the physical attributes of the cameras (e.g., geographical location). This increases the risk of exposing sensitive personal information. (2) ReID tasks may highlight differences between demographic groups. Datasets are often collected from a single region, leading to a lack of diversity in the population represented. This can result in models performing inconsistently across different demographic groups.
> >
> > Our proposed AFM module extracts masks of pedestrians under various scenarios and cameras, followed by clustering of enhanced foreground features. This reduces the risk of leaking background information. Additionally, query images in our approach are selected from the entire dataset without relying on camera IDs, further preventing the exposure of camera locations. Furthermore, training on multiple datasets helps mitigate issues related to demographic diversity and bias. As demonstrated in the experiments addressing Part I Weakness 1, the ICG model exhibits strong generalization capabilities. Thank you again for your valuable suggestions!
> >
> > 4. Weakness 4
> >
> > Thank you for pointing out this issue. We have corrected the formatting of Table 1 and Figure 2 in the revised version. We will carefully review the paper to fix other grammatical or typesetting issues.
> >
> > We sincerely apologize for omitting the additional experimental validations and discussions on ethical considerations in the appendix. These will be included in the appendix of the revised version.
> >
> >
> > 6. Question 1
> >
> > Thanks to your suggestion, we plan to make the code public after the paper is accepted. The github link and detailed description of the code will also be provided in the final version.

---

### Official Review · Reviewer_fRrV · 2024-11-03

**Soundness:** 3
**Presentation:** 3
**Contribution:** 2
**Rating:** 5
**Confidence:** 4

**Summary:**

This paper presents an Instance-level Consistent Graph (ICG) framework for person re-identification, addressing the challenging issues of part misalignment and feature inconsistency. The proposed method integrates attention-based foreground separation, unsupervised human parts clustering, and graph-based structural modeling to achieve instance-level consistency. The framework demonstrates promising results on several benchmark datasets.

**Strengths:**

1 The paper's primary contribution lies in its pragmatic approach to handling part misalignment in person re-ID. Instead of pursuing strict part-level consistency, which often fails under challenging conditions, the authors propose a more flexible instance-level consistency approach.

2 The unsupervised nature of the human parts clustering is particularly noteworthy, as it eliminates the need for additional supervision or pre-trained models, making the solution more deployable in real-world scenarios.

3 The experimental results across multiple datasets demonstrate the effectiveness of this approach.

**Weaknesses:**

1 The primary weakness of this work lies in its limited theoretical novelty and reliance on conventional methodologies. The core components - attention mechanism, K-means clustering, and graph convolutional networks - are well-established techniques that have been extensively studied in the field. While the integration of these components is practical, it does not present significant methodological advancement.

2 The use of basic K-means clustering and standard GCN architecture appears dated compared to recent developments in self-attention mechanisms, advanced clustering techniques, and modern graph learning approaches. The paper would benefit substantially from incorporating more contemporary methodologies and providing stronger theoretical justification for the chosen approach.

3 The paper lacks comprehensive analysis in several crucial aspects. The absence of detailed ablation studies makes it difficult to understand the relative importance of each component. The computational complexity and runtime performance considerations are not adequately addressed, which are crucial factors for practical deployment. The robustness of the clustering approach to different parameters and varying environmental conditions needs more thorough investigation.

**Questions:**

1 Include comprehensive ablation studies and failure case analyses to provide deeper insights into the framework's behavior and limitations. This should be accompanied by detailed computational complexity analysis and runtime performance evaluations.

2 Expand the experimental evaluation to include comparisons with recent transformer-based approaches and demonstrate the method's robustness under various challenging conditions. The addition of qualitative results showing the clustering and graph construction process would enhance the paper's clarity.

---

> ### Author Response · Authors · 2024-11-26
> **Official Response to Reviewer fRrV (PART I)**
>
> We sincerely appreciate your constructive and thoughtful feedback. Below are our supplementary experiments and responses:
>
> 1. Weakness 1
>
> Thank you for your valuable suggestion.
>
> Recent advancements leveraging human parts extraction have shown promise by utilizing discriminative part features. However, the detailed information provided by human parts also presents challenges, particularly in terms of part misalignment. Real-world re-ID scenarios often face issues such as partial occlusion, pose variations, and extreme illumination, which exacerbate misalignment due to limited information about human parts. Despite previous efforts, achieving part-level consistency across all samples remains an ideal but stringent assumption, especially under challenging conditions. Extracting identical features from incomplete information about human parts can lead to a semantic gap between samples of the same instance.
>
> Our ICG framework addresses these challenges by utilizing an attention-based foreground mask to extract pedestrian masks, ensuring that subsequent operations are performed on enhanced foreground features. This approach effectively mitigates the issues of fine-grained human part misalignment and coarse-grained image block misalignment outlined in Figure 1 of our paper. Additionally, pixel-wise human parts clustering enables unsupervised clustering of pedestrian parts. In the PPC module, features from all images of the same ID are clustered together, ensuring that the model maintains instance-level consistency during the clustering process.
>
> Furthermore, the flexible structure graph constructs a dynamic graph for each query image. This graph uses the human parts and their labels obtained via K-means clustering as the initial graph matrix, which is then updated through multiple GCN layers. The GCN strengthens the relationships between correlated human parts and attenuates the effects of misalignment caused by partial occlusion, pose variations, and extreme illumination, thereby forming more robust pedestrian structural features.
>
> While several modules in our framework rely on established methodologies, the proposed approach of leveraging a more flexible instance-level consistency for person re-identification, instead of pursuing strict part-level consistency, proves to be practical and effective. We will include additional experiments and ablation studies to further demonstrate how each component contributes to the overall performance.
>
> 2. Weakness 2
>
> We sincerely thank the reviewer for pointing out the potential limitations of using attention-based foreground mask, basic K-means clustering and a standard graph convolutional network (GCN) architecture in our work. The motivation of our proposed ICG framework, as well as the realization ideas for its components, has been discussed in our response to weakness 1. The ICG framework introduces a more flexible instance-level consistency approach, and our experiments have demonstrated the feasibility and effectiveness of this framework.
>
> However, we acknowledge that incorporating advanced clustering techniques (e.g., spectral clustering, deep embedded clustering), modern self-attention mechanisms, or graph learning approaches may further enhance the methodology. But due to time constraints, we are unable to implement and evaluate these approaches in the current work.
>
> We deeply appreciate the reviewer’s constructive feedback and insightful suggestions. In future work, we plan to explore transformer-based structures and and develop end-to-end instance-consistent re-identification algorithms. Our further research will aim to leverage transformer-based attention mechanism's global dependency, investigate more precise pedestrian part segmentation, and construct refined pedestrian structural graphs.

---

> ### Author Response · Authors · 2024-11-26
> **Official Response to Reviewer fRrV (PART II)**
>
> 3. Weakness 3, Question 1
>
> Thank you to your suggestions. Due to space constraints, we mistakenly omitted some ablation studies. The detailed ablation studies is added in Section 3.3 ablation experiments as follow. The computational complexity analysis could be find in response to Reviewer 7m9c (PART I) Weakness 1 third point.
>
> (1) We further analyse of the AFM performance in the three scenarios shown in Figure 7:
>
> “The effectiveness of the AFM is verified across various scenarios. Furthermore, specific foreground areas exhibit stronger responses, indicating their heightened importance. These areas contain information that better discriminates inter-class differences, thereby facilitating classification tasks. Additionally, the AFM effectively suppresses responses in occluded regions. To a certain extent, it mitigates the impact of background noise, resulting in less noise in human parts.”
>
> (2) We will add an ablation experiment about the number of clustering centers $K$ in PPC with  the following description:
>
> Intuitively, the number of cluster centers, denoted as $K$, determines the granularity of aligned parts when generating part pseudo-labels. In this approach, multiple part regions are obtained by clustering from top to bottom. The larger the value of $K$, the smaller the pixel share of each region, resulting in finer granularity. Consequently, the PPC generates different numbers of confidence maps for the classification of pixel channel features.
>
> In order to explore the influence of the number of clustering centers K on the network performance, the ablation experiment is conducted as the follow table:
>
> | **K** | **Market-1501 (mAP)** | **Market-1501 (Rank-1)** | **DukeMTMC-reID (mAP)** | **DukeMTMC-reID (Rank-1)** | **CUHK03 (mAP)** | **CUHK03 (Rank-1)** | **MSMT17 (mAP)** | **MSMT17 (Rank-1)** |
> |:-----:|:---------------------:|:-----------------------:|:-----------------------:|:-------------------------:|:----------------:|:-------------------:|:----------------:|:-------------------:|
> |   3   |         87.9          |          94.9           |          81.0           |           90.2            |       72.6       |        74.9         |       58.1       |        81.1         |
> |   4   |         87.4          |          94.8           |          81.7           |           90.4            |       72.3       |        74.8         |       58.8       |       **81.7**      |
> |   5   |         87.8          |          94.7           |          81.9           |           91.2            |       72.0       |        75.0         |       58.2       |        80.9         |
> |   6   |       **88.9**        |        **95.4**         |        **82.4**     |         **91.4**     |     **74.4**     |      **76.9**       |     **59.5**     |    81.6         |
> |   7   |         88.5          |          95.3           |          81.8           |           90.8            |       73.6       |        76.6         |       58.5       |        80.9         |
>
> From the results presented in the table, it can be observed that the experiments achieve near-optimal performance at K=6. To approximate real-life scenarios, images often include personal belongings such as backpacks. When the number of clusters is set to K=4, the generated local semantic regions may be relatively accurate, leading to a local optimum. However, when the number of clusters increases to K=7, the granularity of the generated regions becomes too fine, resulting in less effective local features for pedestrians and ultimately degrading network performance. At K=6, personal belongings are identified with the highest probability.
>
> Additionally, we conducted further analysis on the impact of the number of clusters K on the generation of local semantic regions by visualization figure. The visualizations of these experiments are provided in the appendix, comparing the effects of clustering algorithms with various K values against an additional semantic parsing model. As shown, at K=6, potential personal belongings are effectively identified and incorporated into the pedestrian representation. Figure (b) illustrates the parsing results under occlusion conditions. The first two rows show that the PPC module achieves performance nearly comparable to the SCHP algorithm, effectively distinguishing occluded areas. The results in the last row highlight the advantages of our approach in handling occlusion caused by other pedestrians. SCHP struggles to distinguish such cases, treating features of other pedestrians as interference. In contrast, our method clusters features from all images of the same ID, allowing information sharing across instances. This enables our model to discard features of other pedestrians as background, demonstrating the benefits of our approach. Moreover, observing each column in the figure, regardless of the value of K, the PPC module consistently ensures local semantic region alignment across images.

---

> ### Author Response · Authors · 2024-11-26
> **Official Response to Reviewer fRrV (PART III)**
>
> 4. Question 1
>
> Thank you for your valuable suggestions. The comprehensive ablation studies could be find in the above response to Weakness 3. The ICG's behavior and limitations could be find in  find in response to Reviewer 7m9c (PART I) Weakness 2. The computational complexity analysis could be find in response to Reviewer 7m9c (PART I) Weakness 1 third point.
>
> 5. Question 2
>
> * Comparisons with recent transformer-based approaches
>
> As shown in the below Table, our proposed ICG framework achieves competitive or superior results compared to recent transformer-based approaches. ICG attains the best performance with a Rank-1 accuracy and mAP on Market-1501, DukeMTMC-reID and MSMT17 dataseta, but ICG achieves a Rank-1 accuracy of  76.9% and mAP of  77.4% on CUHK03 dataset, getting a lower result compared with AAformer. AAformer introduces an auto-aligned transformer that automatically locates both human and non-human parts at the patch level. Its self-attention mechanism models global dependencies and ensures fine-grained patch alignment. This may be the reason why ICG's performance is slightly worse than AAformer on some datasets.
>
> | Algorithm   | Venue   | Market-1501 (Rank-1) | Market-1501 (mAP) | DukeMTMC-reID (Rank-1) | DukeMTMC-reID (mAP) | MSMT17 (Rank-1) | MSMT17 (mAP) | CUHK03 (Rank-1) | CUHK03 (mAP) |
> |:-----------:|:-------:|:--------------------:|:-----------------:|:----------------------:|:-------------------:|:---------------:|:------------:|:---------------:|:------------:|
> | TransReID   | ICCV21  |        95.2         |       **88.9**       |         90.6          |        82.2         |      -      |     -     |        -        |      -       |
> | AAformer    | TNNLS23 |        **95.4**         |        88.0         |         90.1          |        -          |      -        |     -     |     **77.6**    |   **74.8**  |
> | DAAT        | IVC23   |        95.1         |       88.8        |         90.6          |         82.0      |        -        |      -       |        -        |      -       |
> | ICG (Ours)  |    -    |      **95.4**       |       **88.9**     |       **91.4**        |      **82.4**        |    **81.6**     |  **59.5**    |      76.9       |     74.4     |
>
> * Qualitative results showing the clustering and graph construction process
>
> Some of the qualitative results are mentioned in the above response. Furthermore, in order to explore the variation of pseudo-labels in local semantic regions with training, we visualized  the pseudo-label change process of each local semantic region when K is 6. More figures and explanations of the experimental results can be referred to the appendix of revised paper that we will upload later.

---

> > ### Comment · Reviewer_fRrV · 2024-12-02
> >
> > I have thoroughly reviewed the revised manuscript and carefully considered all aspects, including the overall quality of the paper, the feedback from other reviewers, and the current state of research in this field. After comprehensive evaluation, I maintain my score of 5 points. The primary reason for this decision is that the manuscript's innovation level and contribution to the field remain limited.

---

> > > ### Author Response · Authors · 2024-12-03
> > > **Thanks for the further comments.**
> > >
> > > Thank you for taking the time to thoroughly review our manuscript and providing valuable feedback. We fully understand your expectations regarding the level of innovation and contribution to the field, and your comments are highly instructive for guiding the improvement of our future work.

---

### Official Review · Reviewer_xMUt · 2024-11-05

**Soundness:** 2
**Presentation:** 3
**Contribution:** 3
**Rating:** 5
**Confidence:** 4

**Summary:**

This paper proposes a new person reID method, including two main parts, i.e., the attention-based foreground mask unit and the unsupervised clustering unit. Futhermore, a graph model is built for instance-level consistence. The experimental results are good.

**Strengths:**

1. The motivation of address the confilct between the fine-grained and coarse-grained pipelines are reasonable, and the proposed graph model for instance consistency is new.
2. The introduced pixel-wise human parts clustering is novel, which play an important role for balance the fine-or-coarse constrain for parts.
3. The experimental evaluation is sufficient and the results are excellent.

**Weaknesses:**

1. The attention-based foreground mask learning is not new, which has been propsoed in previous work [1] for person reID.  The difference or the advantage of the the proposed AFM should be discussed.
2. The compared methods are most out-of-date, more recently propsoed methods should be compared.


[1] Mask-guided Contrastive Attention Model for Person Re-Identification. CVPR.2018

**Questions:**

Why not adopt the well-segmented foreground mask directly? The learned attention map involves lots of noises.

---

> ### Author Response · Authors · 2024-11-25
> **Official Response to Reviewer  xMUt**
>
> We sincerely appreciate your constructive and thoughtful feedback. Below are our responses and detailed explanation:
> 1. Weakness 1
>
> The proposed attention-based foreground mask (AFM) learning method differs from approaches that rely on additional semantic information, such as skeleton poses, human part segmentation, or bounding boxes. These methods typically involve pre-trained detection or parsing models, which increase model complexity. It also contrasts with coarse partitioning methods for pedestrian images, which often lead to part misalignment issues.
>
> In prior work [1], researchers utilized RGB-Mask pairs as inputs to learn features from the body and background regions separately for contrastive learning. This method first employs a pre-trained pedestrian segmentation model to generate the pedestrian mask, which is then combined with the RGB image as input. Each image is augmented with the mask as prior information, and pedestrian re-identification features are learned through three main streams (the full-stream, the body-stream, and the background-stream).
>
> Our method differs with [1] in several key aspects. First, we rely solely on the original dataset for training, with no need for additional data as model input. Second, the AFM module is based on the observation that the foreground response in feature maps tends to be larger than the background response. The spatial attention layer enhances the contrast between the foreground and background by increasing the attention values of foreground pixels. Through classification loss and part-parsing loss, the network is progressively guided to focus more on foreground features during learning. Moreover, AFM selectively enhances the foreground and suppresses the background at feature-level. In contrast, features extracted from image-level foreground mask (by [1]) may include noise at the mask edges due to transitional areas. Our method effectively avoids this limitation, improving the robustness of the learned features.
>
> 2. Weakness 2
>
> Thank you for your valuable suggestion. We removed certain outdated methods from our comparison updated some recent methods. A more complete comparison table will be given in the revised version of the paper. As shown in the below Table, our proposed ICG framework achieves competitive or superior results compared to these recent approaches. ICG attains the best performance with a Rank-1 accuracy and mAP on DukeMTMC-reID and MSMT17 dataset, and ICG achieves a Rank-1 accuracy of 95.4% and mAP of 88.9% on Market-1501, matching or surpassing AAformer and MSINET.
>
> MSINet employs a multi-scale interaction search strategy, significantly enhancing the model's discriminative power through contrastive learning of objects. Our ICG framework did not use multi-scale feature extraction technology, which may lead to the loss of some details captured by the network. AAformer introduces an auto-aligned transformer that automatically locates both human and non-human parts at the patch level. Its self-attention mechanism models global dependencies and ensures fine-grained patch alignment. This may be the reason why ICG's performance is slightly worse than AAformer on some datasets.
>
> | Algorithm   | Venue   | Market-1501 (Rank-1) | Market-1501 (mAP) | DukeMTMC-reID (Rank-1) | DukeMTMC-reID (mAP) | MSMT17 (Rank-1) | MSMT17 (mAP) | CUHK03 (Rank-1) | CUHK03 (mAP) |
> |:-----------:|:-------:|:--------------------:|:-----------------:|:----------------------:|:-------------------:|:---------------:|:------------:|:---------------:|:------------:|
> | BPBReID     | WACV23  |        95.1         |        87.0         |         89.6          |        78.3         |        -        |      -       |        -        |      -       |
> | MSINET      | CVPR23  |        95.3         |       **89.6**        |          -            |          -          |      80.7       |     **59.5**     |        -        |      -       |
> | TransReID   | ICCV21  |        95.2         |       88.9        |         90.6          |        82.2         |      -      |     -     |        -        |      -       |
> | AAformer    | TNNLS23 |        **95.4**         |        88.0         |         90.1          |        -          |      -        |     -     |     **77.6**    |   **74.8**  |
> | DAAT        | IVC23   |        95.1         |       88.8        |         90.6          |         82.0      |        -        |      -       |        -        |      -       |
> | ICG (Ours)  |    -    |      **95.4**       |       88.9     |       **91.4**        |      **82.4**        |    **81.6**     |  **59.5**    |      76.9       |     74.4     |

---

### Note · Authors · 2025-01-24

I have read and agree with the venue's withdrawal policy on behalf of myself and my co-authors.